# Research on Identification and Detection of Transmission Line Insulator Defects Based on a Lightweight YOLOv5 Network

**Zhilong Yu** [1], **Yanqiao Lei** [1,*], **Feng Shen** [2], **Shuai Zhou** [3] and **Yue Yuan** [2]

1 College of Automation, Harbin University of Science and Technology, Harbin 150080, China; zlyu@hrbust.edu.cn
2 School of Instrumentation Science and Engineering, Harbin Institute of Technology, Harbin 150001, China; fshen@hit.edu.cn (F.S.); 18b901010@stu.hit.edu.cn (Y.Y.)
3 Electric Power Research Institute, Yunnan Power Grid Co., Ltd., Kunming 650217, China; zhoushuailijinmei@163.com
* Correspondence: lei820029589@163.com

**Abstract:** Transmission line fault detection using drones provides real-time assessment of the operational status of transmission equipment, and therefore it has immense importance in ensuring stable functioning of the transmission lines. Currently, identification of transmission line equipment relies predominantly on manual inspections that are susceptible to the influence of natural surroundings, resulting in sluggishness and a high rate of false detections. In view of this, in this study, we propose an insulator defect recognition algorithm based on a YOLOv5 model with a new lightweight network as the backbone network, combining noise reduction and target detection. First, we propose a new noise reduction algorithm, i.e., the adaptive neighborhood-weighted median filtering (NW-AMF) algorithm. This algorithm employs a weighted summation technique to determine the median value of the pixel point's neighborhood, effectively filtering out noise from the captured aerial images. Consequently, this approach significantly mitigates the adverse effects of varying noise levels on target detection. Subsequently, the RepVGG lightweight network structure is improved to the newly proposed lightweight structure called RcpVGG-YOLOv5. This structure facilitates single-branch inference, multi-branch training, and branch normalization, thereby improving the quantization performance while simultaneously striking a balance between target detection accuracy and speed. Furthermore, we propose a new loss function, i.e., Focal EIOU, to replace the original CIOU loss function. This optimization incorporates a penalty on the edge length of the target frame, which improves the contribution of the high-quality target gradient. This modification effectively addresses the issue of imbalanced positive and negative samples for small targets, suppresses background positive samples, and ultimately enhances the accuracy of detection. Finally, to align more closely with real-world engineering applications, the dataset utilized in this study consists of machine patrol images captured by the Unmanned Aerial Systems (UAS) of the Yunnan Power Supply Bureau Company. The experimental findings demonstrate that the proposed algorithm yields notable improvements in accuracy and inference speed compared to YOLOv5s, YOLOv7, and YOLOv8. Specifically, the improved algorithm achieves a 3.7% increase in accuracy and a 48.2% enhancement in inference speed compared to those of YOLOv5s. Similarly, it achieves a 2.7% accuracy improvement and a 33.5% increase in inference speed compared to those of YOLOv7, as well as a 1.5% accuracy enhancement and a 13.1% improvement in inference speed compared to those of YOLOv8. These results validate the effectiveness of the proposed algorithm through ablation experiments. Consequently, the method presented in this paper exhibits practical applicability in the detection of aerial images of transmission lines within complex environments. In future research endeavors, it is recommended to continue collecting aerial images for continuous iterative training, to optimize the model further, and to conduct in-depth investigations into the challenges associated with detecting small targets. Such endeavors hold significant importance for the advancement of transmission line detection.

**Keywords:** insulators; defect detection; YOLOv5; noise reduction; RepVGG

## 1. Introduction

Transmission line serve as the primary means of transportation within the national power system, thereby, playing a crucial role in ensuring the system's stable operation. In light of rapid advancements in drone inspection technology and artificial intelligence [1], the inspection of transmission lines in China has only achieved a level of intelligence [2] whereby preset flight paths enable automatic aerial photography [3,4]. Nevertheless, the processing of these aerial images still relies on manual screening, which results in slow detection speeds and substantial consumption of human and material resources. Consequently, intelligent detection and recognition of defects capabilities within vast quantities of transmission line images hold immense significance.

Extensive research has been conducted both domestically and internationally on the detection of defects in transmission lines, yielding phased achievements [5–7]. Notably, algorithms based on convolutional neural networks have demonstrated remarkable performance in target detection [8] and project defect detection [9]. In one study [10], the authors proposed a method that employed regional divisions of high-capacity convolutional neural networks, known as R-CNN, to locate and segment detection objects. Subsequently, faster models were developed, such as Fast R-CNN [11] and Faster R-CNN [12]. Another study [13] optimized the R-CNN approach by enhancing regional divisions and employing neural network architecture search technology, resulting in promising outcomes in fabric defect detection. Additionally, a modified Faster R-CNN algorithm was proposed in a separate study [14], which utilized a cluster algorithm for the initial test box and improved the loss function to address the imbalance of positive and negative samples. Furthermore, the authors of another publication [15] employed a combination of an empty-hole convolution network and a deep Q network for feature extraction, while incorporating an attention mechanism module to model the defects. The proposed approach exhibited commendable robustness. In a separate study [16], a TensorFlow platform was introduced, along with a Faster R-CNN model based on this platform. The Concept-Resnet-v2 network was employed as the fundamental feature extraction network, enabling network structure adjustments and parameter optimizations. Considering the current development trend, the R-CNN algorithm is evolving towards lightweight implementations. However, the inherent limitations of this two-stage algorithm, such as its relatively complex model and slower reasoning speed, make it unsuitable for projects requiring large picture batches and limited hardware resources, particularly for real-time monitoring.

The current target recognition algorithms, in addition to the two-stage algorithm led by R-CNN, is the single-stage algorithm dominated by YOLO [17]. This single-stage algorithm model is relatively concise, with faster inference, which means that the practical application of memory occupation is relatively low; it has better generalization for real-time detection of engineering scenarios and has been applied to power transmission lines [18]. In another publication [19], a combination of YOLOv3 and Faster CNN was applied to inspect transmission lines, effectively detecting and determining the types and locations of defects. Despite the notable accuracy breakthrough, the complexity of the model renders it unsuitable for practical project applications. Moreover, it consumes a significant amount of memory and hampers reasoning speed. In a different study [20], the author enhanced the YOLO network structure by integrating a residual network with a convolutional network. This modification led to substantial improvements in both accuracy and reasoning speed. However, it is worth noting that there is still a risk of insulation being overlooked during the convolution process, resulting in a relatively low degree of adaptation.

In the domain of lightweight networks, such as the Ghost-Net [21] series, Mobilenetv1 [22] series, and shufflenet [23] series, significant advancements have been made. However, it has been observed in the literature [24] that deep separable convolution employs a substantial number of low FLOPs and entails high data read/write operations. This characteristic renders it more compatible with CPU and ARM mobile devices, while proving less efficient for hardware with high parallelism, such as GPUs. Given that the transmission branch of aerial image screening relies on GPU hardware, RepVGG [25], a VGG-like convolutional

architecture, is better suited to harness the computational power of GPUs. By employing distinct network structures for inference and training, RepVGG achieves a harmonious balance between accuracy and speed, making it an ideal foundational network for this study. Nevertheless, when applied to machine inspection images, the detection of small target insulators still exhibits a high rate of leakage and false positives, necessitating further improvement. Furthermore, the current algorithm overlooks the influence of image quality on detection in practical applications. Therefore, it remains crucial to prioritize the construction of an image preprocessing network while enhancing the algorithm network.

Within the realm of YOLO series algorithms, YOLOv5 stands out as the most prominent and widely adopted approach. Extensive research [26] has been conducted to enhance the detection of small target defects in integrated circuits, primarily by augmenting the YOLOv5 detection component and incorporating the SE layer. Another study [27] introduced the BottleneckCSP module into the primary network for header detection, while leveraging Ghost convolution to reduce model complexity and strike a favorable balance between accuracy and speed. This research has yielded valuable insights into the design of lightweight models. In the context of industrial testing, a separate investigation [28] focused on constructing a lightweight backbone network and substituting SPP with experience-driven wild modules. This approach not only improves accuracy while maintaining model efficiency, but also offers practical guidance for engineering deployment. However, it is important to note that the actual deployment scenario was not fully considered in this study. The detection of insulation poses a significant challenge due to the substantial imbalance between positive and negative samples, leaving ample room for improvement. Recent efforts have focused on addressing common issues encountered in the YOLO algorithm through the utilization of artificially curated datasets. However, it is important to note that the quality of images obtained through direct transmission from drones may not always be guaranteed, and the power grid dataset remains confidential. Furthermore, the performances of most algorithms have not been thoroughly validated using actual machine patrol images. Consequently, there is still considerable potential for enhancing the YOLO algorithm's ability to detect target defects in real-world scenarios.

Line inspection defect reports of power companies reveal that the majority of defective images are associated with insulators. This finding underscores the practical relevance of employing YOLOv5 for insulator defect detection in engineering applications.

(1) In the current landscape of computer models, the majority of existing models tend to be complex and pose challenges regarding their practical implementation in typical configurations. In this paper, we address this problem by reconfiguring the RepVGG network structure and propose a novel primary network structure called RcpVGG (reconstitution VGG) as the backbone of YOLOv5. The aim of this reconstruction is to streamline the reasoning architecture, enabling faster inference, improving the accuracy of small target detection, and enhancing the overall applicability of the algorithm in various projects.

(2) Improving the direct acquisition of drone imagery is the focus of this study, given the complexity of the mechanical vibration background and the presence of attached images. In this paper, the filtering method of the adaptive median filtering algorithm is improved, and a new filtering algorithm is proposed, i.e., NW-AMF, with the aim to mitigate the impact of image quality problems on the accuracy of image defect detection, so as to improve the noise reduction effect and detection accuracy.

(3) In order to cope with the challenges of small targets and unbalanced positive samples on target detection, we propose a new loss function, i.e., Focal EIOU. Firstly, the original CIOU loss function is replaced by the EIOU loss function with better convergence effect, and is combined with the Focal loss function to suppress positive samples effectively. Together, these improvements contribute by improving the accuracy and acceleration of convergence speed.

(4) To verify the efficacy of the proposed algorithm, we use a completely new dataset. This study employs a dataset comprised of non-public machine patrol images obtained

from the drone system of a power supply bureau company. This dataset is utilized for both training and validation purposes, serving as a means to assess the practical applicability of the algorithm.

## 2. YOLOv5 Network

### 2.1. YOLOv5 Algorithm

Figure 1 illustrates the network architecture of YOLOv5, which incorporates mosaic data augmentation and adaptive pin-frame computation techniques to enrich the dataset and obtain an optimal size suitable for the dataset. The backbone network primarily consists of CBS (Conv + BatchNorm + SiLU), C3, and SPPF modules. During the feature extraction stage, the preprocessed image serves as input to the backbone network, which extracts features from various layers corresponding to the target being detected. These extracted features are subsequently fused in the feature fusion network. The Neck component of YOLOv5 is responsible for integrating features from different scales and levels. The Head component performs multi-scale detection on feature maps of varying sizes, ultimately yielding the coordinate positions and sizes of the target boxes.

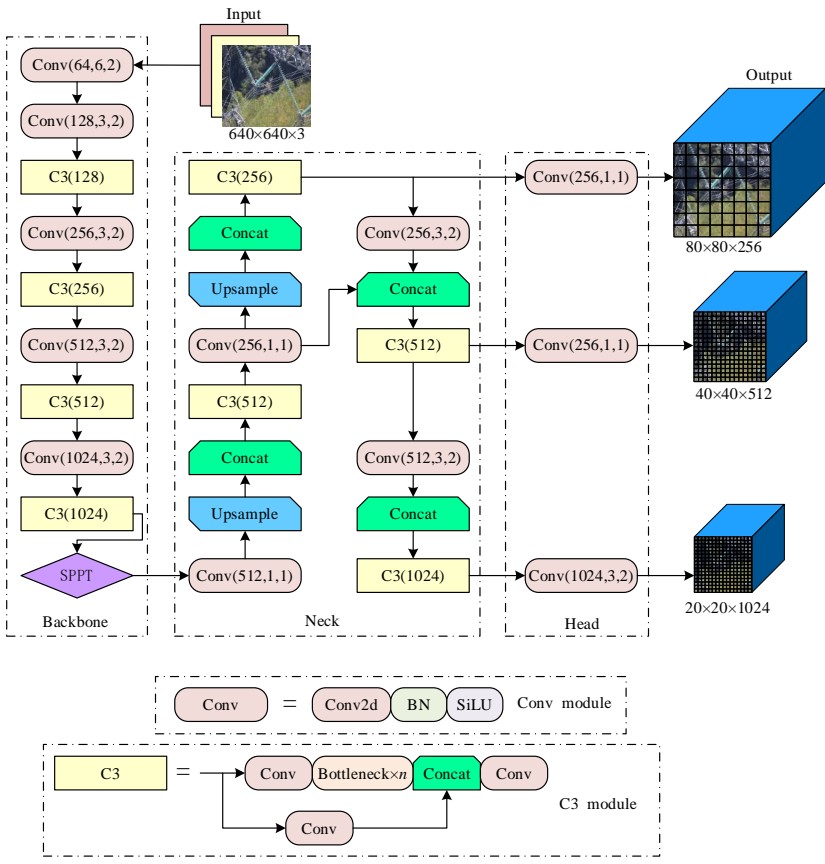

**Figure 1.** YOLOv5 network structure diagram.

### 2.2. The Heavy Parameters of the Main Network

Despite the continuous emergence of algorithms for complex branch structures and the increasing accuracy of detection, the industry still extensively employs algorithms based on the VGG network. This is primarily due to the VGG network's fast reasoning speed, facilitated by its single-road structure, which eliminates the need for additional branches. Moreover, the VGG network reduces memory consumption and offers a relatively high level of flexibility in its single-way architecture, enabling improvement at each layer. However, these advantages are accompanied by evident shortcomings, such as poor performance and relatively low detection accuracy, resulting in performance deviations.

According to the current algorithm network structure and performance, it can be concluded that incorporating a multi-branch architecture during the training process is beneficial for improving accuracy. However, during the inference process, the use of the multi-branch architecture can impact reasoning speed, which may be insufficient for achieving faster reasoning. To address this limitation, a novel network architecture called RepVGG was proposed in the literature for the first time [18]. The RepVGG training process employs a multi-branch structure, consisting of $3 \times 3$ Conv, $1 \times 1$ Conv, and residual branches, which enhances discoverability. In contrast, the reasoning process adopts a single-road minimalist architecture, significantly boosting inference speed. Refer to Figure 2 for a visual representation of this architecture.

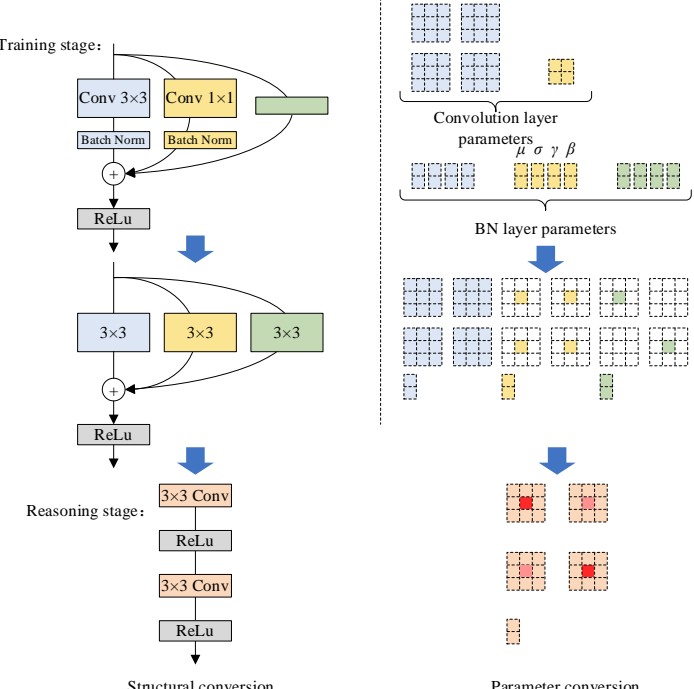

**Figure 2.** RepVGG main network diagram.

The following can be observed through the constructive diagram. For each branch of the convolutional layer, the method of combining with the Batch Norm layer is used in $C_1$, $C_2$ and the input and output channel of the $3 \times 3$ convolutional layers can be represented by $W^{(3)} \in R^{C_1 \times C_2 \times 3 \times 3}$. Similarly, the $1 \times 1$ convolutional layers can be expressed as $W^{(1)} \in R^{C_2 \times C_1}$ and $W$ stands for $3 \times 3$ convolution kernel. The accumulated average, standard deviation, and proportional factors are combined with the three branches and the Batch Norm layer. The deviation is $\mu^{(3)}, \beta^{(3)}, \sigma^{(3)}, \gamma^{(3)}, \mu^{(1)}, \sigma^{(1)}, \gamma^{(1)}, \beta^{(1)}, \mu^{(2)}, \beta^{(2)}, \sigma^{(2)}, \gamma^{(2)}$. The convolutional process is represented by *. When the input and output image variables are the same. When satisfying $C_1 = C_2, H_1 = H_2, W_1 = W_2$, then:

$$\begin{aligned}
M^{(2)} = \ & bn\left(M^{(1)} * W^{(3)}, \mu^{(3)}, \sigma^{(3)}, \gamma^{(3)}, \beta^{(3)}\right) \\
& + bn\left(M^{(1)} * W^{(1)}, \mu^{(1)}, \sigma^{(1)}, \gamma^{(1)}, \beta^{(1)}\right) \\
& + bn\left(M^{(1)}, \mu^{(0)}, \sigma^{(0)}, \gamma^{(0)}, \beta^{(0)}\right)
\end{aligned} \tag{1}$$

Suppose only the $3 \times 3$ convolution and $1 \times 1$ convolutional branches are considered. In this case, the reasoning function BN is expressed as:

$$bn(M, \mu, \sigma, \gamma, \beta)_{i,i,::} = (M_{:,i,:,:} - \mu_i)\frac{\gamma_i}{\sigma_i} + \beta_i, \forall 1 \le i \le C_2 \tag{2}$$

Use $W'$ and $b'$ to indicate the convolution nuclear and bias vector coefficients obtained after the BN layer, then there are:

$$W'_{i,i,,:} = \frac{\gamma_i}{\sigma_i} W_{i,\cdot,:,:}, \ b'_i = -\frac{\mu_i \gamma_i}{\sigma_i} + \beta_i \tag{3}$$

$$\text{Available}: \ \text{bn}(M * W, \mu, \sigma, \gamma, \beta)_{:,i,i,:} = (M^*W')_{i,i,,:} + b'_i, \ \forall 1 \le i \le C_2 \tag{4}$$

The remaining identity branches can be treated as a volume layer with a weight of $1 \times 1$, allowing for the application of conversion. By incorporating biases into the three branches, the final bias weight can be obtained. This module can be considered to be a convolution module, and the reasoning process can be viewed as a single-road architecture stacked using this $3 \times 3$ convolution module.

## 3. RcpVGG-YOLOv5 Algorithm

In the context of insulator defect detection, utilization of the YOLOv5 has revealed a notable drawback in terms of its sluggish inference speed. Additionally, it suffers from a high rate of misunderstanding, resulting in subpar accuracy. To address these issues, the present study proposes a novel approach that leverages RepVGG as the primary network prototype to reconfigure the existing framework of YOLOv5. The resultant algorithm, denoted as RcpVGG-YOLOv5, integrates noise and lightweight target detection techniques, as visually depicted in Figure 3.

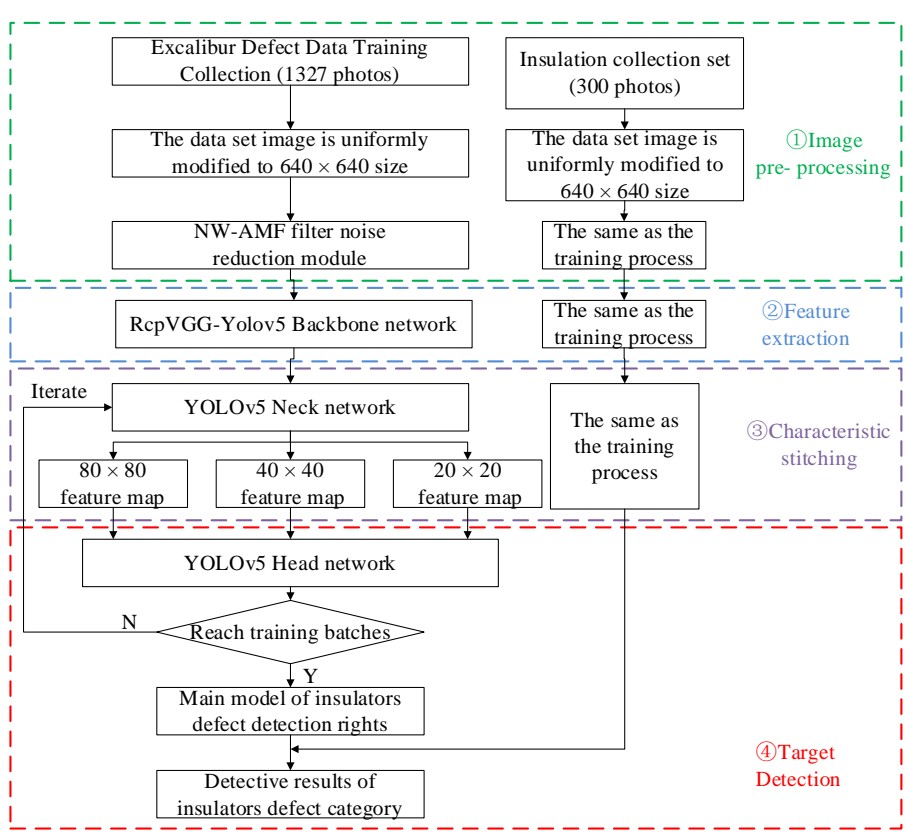

**Figure 3.** Overall algorithm flowchart.

### 3.1. Improved Adaptive Median Filter Algorithm NW-AMF Filtering

When drone footage is captured in a natural environment along a predetermined flight path, the resulting images may be impacted by various types of noise, which include noise from the complex background environment, as well as mechanical jitter and current within the drone itself. Such noise can negatively affect subsequent target testing and other

analytical processes. As a result, preprocessing of noise reduction in the input image is necessary [29]. Adaptive median filtering is the main method of traditional noise reduction filtering [30]. The principle is: Set two processes, i.e., A and B. The pixels corresponding to the $(i, j)$ pixels at the coordinate are $X(i, j)$, and the maximum size corresponding to the corresponding window is $M_{max}$. Set $Z_{max}$, $Z_{min}$, $I_{med}$, which are the maximum, minimum value, and $Z(i, j)$ median value of the corresponding window gray, respectively. The two processes A and B meet the formulas:

$$Z_{A1} = I_{med} - Z_{min} \tag{5}$$

$$Z_{A2} = Z_{max} - I_{med} \tag{6}$$

$$Z_{B1} = Z_{(i,j)} - Z_{min} \tag{7}$$

$$Z_{B2} = Z_{max} - Z_{(i,j)} \tag{8}$$

Upon transmitting the noisy image to the filter network, the first step involves conducting grayscale extraction to ascertain whether it falls within the median range. This process is governed by Equations (5) and (6). Subsequently, the conditions are evaluated to determine if they are satisfied, i.e., $Z_{A1} > 0$ and $Z_{A2} > 0$. Following this, an assessment is made to verify if the grayscale value of the set window meets the criteria, i.e., $Z_{B1} > 0$ and $Z_{B2} > 0$. If these conditions are met, it can be concluded that the pixels are non-noisy, and the actual grayscale value, denoted as $Z(i, j)$, is outputted. Conversely, if the conditions are not met, the output will be the median grayscale value, represented as $I_{med}$.

However, in the context of comprehensive high-definition image analysis, the impact on the window is significant once the image details reach a relatively complete state. Optimal information performance of the image details is achieved when the window value is set to a smaller magnitude; however, this comes at the expense of compromised noise reduction capabilities. Conversely, a larger window value enhances filtering performance, albeit at the risk of inducing excessive blurring in the image. To address this issue, the present study introduces an enhanced noise reduction method, namely the adaptive neighborhood-weighted median filtering (NW-AMF) method. The neighboring domain of $Z(i, j)$ is expressed as shown in Figure 4.

**Figure 4.** Neighboring domain indication diagram.

Let the set of corresponding neighborhood pixel values be denoted as $f_n(i, j)$, and let the set of neighborhood pixel values in the up, down, left, and right directions be denoted as $W[f(i, j)]$. If the $f_n(i, j)$ value satisfies the conditions of being either 0 or 255, it is considered to be noise and subsequently eliminated. The remaining set of neighborhood pixel values is then used to calculate the median value, denoted as $\text{Med}(W[f(i, j)])$. Weighting coefficients are assigned to each pixel according to Equations (9) and (10). Finally, the remaining pixels

and their corresponding weighting coefficients are multiplied and summed to obtain the final filtering output, as illustrated in Equation (11):

$$\text{sum} = \sum_{n=1}^{N} \frac{1}{1 + (f_n(i,j) - \text{Med}\{W[f(i,j)]\})^2} \tag{9}$$

$$w_n(i,j) = \frac{1}{\left(1 + [f_n(i,j) - \text{Med}\{W[f(i,j)]\}]^2\right) \times \text{sum}} \tag{10}$$

$$f(i,j) = \sum_{n=1}^{N} f_n(i,j) \times w_n(i,j) \tag{11}$$

Among these variables, $N$ represents the total number of pixels in $W[f(i,j)]$. Once the neighboring pixel collection undergoes the filtering process, the size of the weighted coefficient for each pixel point $w_n(i,j)$ is obtained. The resulting filter output is represented by $f(i,j)$. The overall structure of this process can be summarized as in Figure 5:

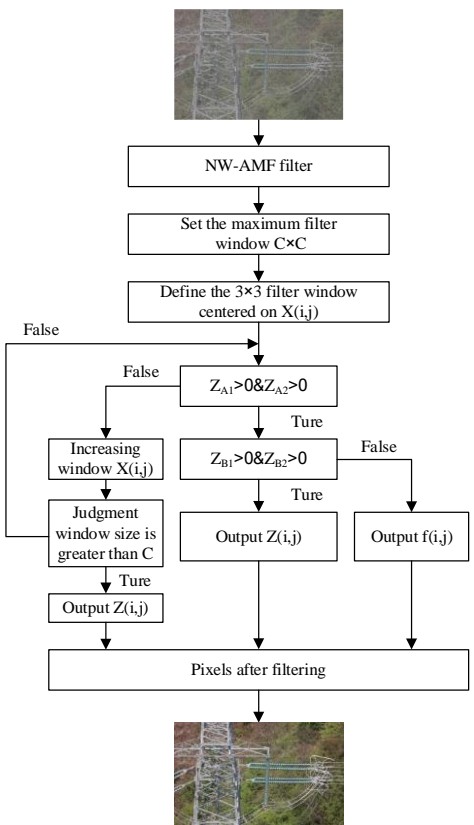

**Figure 5.** Improved adaptive filtering noise reduction method flow diagram.

## 3.2. Reconstruction of the RepVGG Main Network: RcpVGG

The RepVGG network exhibits a notable enhancement in terms of inference speed, while simultaneously striking a balance between performance and accuracy, thereby fulfilling the fundamental requirements for target detection. However, it has been observed that the accuracy of RepVGG experiences a substantial decline. Furthermore, the rate of misinterpretation escalates, rendering the network's performance unsuitable for direct deployment. In light of these circumstances, here, we introduce a novel approach, namely RcpVGG (reconstitution VGG), with the aim of augmenting its detection capabilities while preserving its commendable reasoning prowess.

Upon deploying RepVGG in conjunction with YOLO, a notable deterioration in performance and a collapse in quantization emerge as pressing concerns. Notably, the distribution of input channel weights in the model, as well as the tensor distribution values of the output channel, exhibit favorable characteristics that aid in mitigating quantization errors. In an effort to investigate the quantization error, the study referenced as [31] conducted ablation experiments on the RepVGG architecture, thereby quantifying the error associated with each branch. Intriguingly, it was discovered that the input error experienced a significant increase subsequent to traversing the $1 \times 1$ branches and identity branches of the BN layer. Building upon this revelation, this study adopts a similar approach by eliminating the $1 \times 1$ and identity branches. To address the issue of variance drift, the three branches are subsequently aggregated, and to ensure stability during the training process, a BN layer is introduced after the summation of the three branches. For further details regarding the parameters, please refer to Section 3.1. Consequently, the reconstructed model is represented by Equation (12):

$$M_{(2)} = M_{(1)} * \left[ \sum_{k \in \{0,1,3\}} \text{Reshape}\left( \frac{\gamma_{(k)}}{\sqrt{\epsilon + \sigma_{(k)} * \sigma_{(k)}}} \right) * W_{(k)} \right] + \sum_{k \in \{0,1,3\}} \left[ \beta_{(k)} - \frac{\gamma_{(k)}^* \mu_{(k)}}{\sqrt{\epsilon + \sigma_{(k)} * \sigma_{(k)}}} \right] \quad (12)$$

Reshape() corresponds to the convolutional dimension with the underlying structure. $M_{(1)} \in R^{N \times C_1 \times H_1 \times W_1}$ and $M_{(2)} \in R^{N \times C_2 \times H_2 \times W_2}$ represent the inputs and outputs, respectively. The cumulative mean, standard deviation, scale factor, and deviation of the three branches, after their amalgamation with the BN layer during training, are represented by $\mu$, $\gamma$, $\beta$, and $\sigma$, respectively. Following the concatenation of the three branches, the Batch Norm layer is applied to facilitate the reasoning function. The architectural depiction of this structure can be observed in Figure 6.

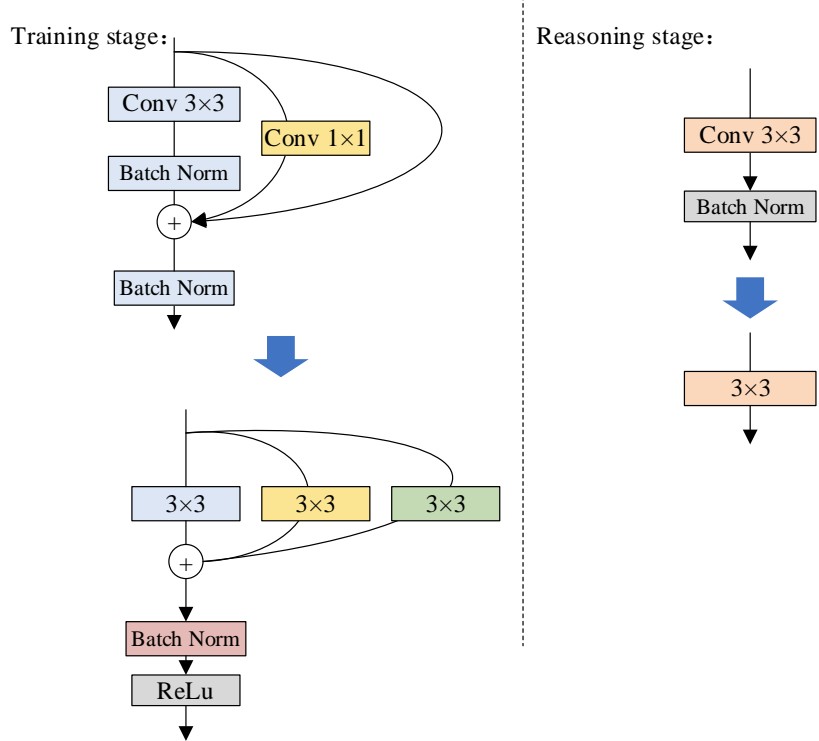

**Figure 6.** The improved RcpVGG network structure diagram.

### 3.3. Improvement of Loss Function

In YOLOv5, the estimation of dissimilarity between the predicted values of the network model and the actual values is commonly accomplished through the utilization of

the loss function. This loss function serves as a crucial metric for evaluating the algorithm network's efficacy. Among the fundamental metrics employed, the metric intersection over union (IOU) stands out. Its formula is as follows:

$$IOU = \frac{A \cap B}{A \cup B} \tag{13}$$

where *A* and *B* represent the area of the prediction box and the real frame; however, the IOU loss defects are relatively evident. It solely focuses on measuring overlap and fails to account for factors such as overlapping methods, shapes, and colors. To address this issue, the GIOU loss function was introduced. It seeks to rectify the drawback by assigning greater importance to the boundary frame. However, the GIOU loss function suffers from algorithm degeneration and slow convergence due to its computationally intensive nature [32]. In an attempt to enhance detection accuracy, the DIOU loss function incorporates the consideration of the distance between the target box and the prediction box, as well as the overlapping rate and scale. Nevertheless, it overlooks the vertical ratio of the return of the target box [33].

The CIOU loss function is employed in YOLOv5, and its calculation formula is as follows:

$$\mathcal{R}_{CIOU} = \frac{\rho^2(b, b^{\text{gt}})}{c^2} + \alpha v \tag{14}$$

$$v = \frac{4}{\pi^2}\left(\arctan\frac{w^{gt}}{h^{gt}} - \arctan\frac{w}{h}\right)^2 \tag{15}$$

$$\alpha = \frac{v}{(1 - IOU) + v} \tag{16}$$

$$\mathcal{L}_{CIOU} = 1 - IoU + \frac{\rho^2(b, b^{\text{gt}})}{c^2} + \alpha v \tag{17}$$

In Formula (14), the center points of the target and prediction frames are denoted as *b* and $b^{\text{gt}}$, respectively. The widths of the target and prediction frames are represented by *w* and $w^{gt}$, while the heights are denoted as and *h* and $h^{gt}$, respectively. The European-style distance between the center point of the prediction box and the real box is represented by $\rho$. Additionally, *c* represents the diagonal distance that encompasses the minimum enclosing area, including both the prediction box and the real frame. The parameter *v* quantifies the consistency of the length and width ratio, while $\alpha$ denotes a weighting parameter. The types (15) and (16) are derived from Xa, and they are used in the final calculation of the loss function, as shown in Formula (17). The CIOU [34] introduces the vertical and horizontal ratios of the target box to the prediction box by increasing the influencing factor. However, due to the discrepancy between the reference value and the relative value, the difference in the number of positive and negative samples for small targets is significant.

Accordingly, in this article, we suggest adoption of the EIOU loss function, which offers several advantages. Instead of relying on the coordinate ratio, it utilizes the discrepancy in horizontal and vertical coordinates, resulting in enhanced convergence speed. Moreover, this approach provides a more precise depiction of the box. The expression for the loss function is given by Formula (18):

$$\begin{aligned}\mathcal{L}_{\text{EIoU}} &= \mathcal{L}_{\text{IoU}} + \mathcal{L}_{\text{dis}} + \mathcal{L}_{\text{asp}}\\ &= 1 - IOU + \frac{\rho^2(b, b^{\text{gt}})}{C^2} + \frac{\rho^2(w, w^{\text{gt}})}{C_w^2} + \frac{\rho(h, h^{\text{gt}})}{C_h^2}\end{aligned} \tag{18}$$

The EIOU loss function [35] encompasses three distinct components to calculate functional losses. These components include the overlap loss functions for the predictive and real boxes, the central distance loss functions, and the horizontal and vertical loss functions. These components are denoted as $\mathcal{L}_{\text{IoU}}$, $\mathcal{L}_{\text{dis}}$, and $\mathcal{L}_{\text{asp}}$, respectively. In this

context, $b, b^{gt}$ represents the central point shared by the two boxes, while $\rho$ signifies the European-style distance between the two center points. In addition, $c$ represents the diagonal distance of the intersecting region between the two frames, while $C_w$ and $C_h$ correspond to the minimum width and height of the two frames, respectively.

Simultaneously, the Focal loss function [36] is introduced as a means to address the issue of the significant disparity in quality between positive and negative samples in practical engineering scenarios. This is expressed by Equation (19) as follows:

$$FocalLoss = -\alpha_t(1-p_t)^{\gamma}\log(p_t) \tag{19}$$

where $\alpha_t$ represents the weighted coefficient of positive and negative samples, which can change the degree of contribution of the positive sample to the loss function by regulating the parameter and $p_t$ denotes the difficulty of controlling the sample classification. The $\gamma$ value is used to adjust the weighted coefficient of the parameter $p_t$.

According to the literature [37], occurrence of the gradient anomaly can adversely affect the experimental process when the target frame is significantly smaller than the image scale. In order to mitigate this issue, the present study incorporates the concept of the Focal loss function to enhance the EIOU loss function in terms of gradient direction. This modification aims to minimize the influence of low-quality samples on the gradient, as demonstrated by Formula (20):

$$\mathcal{L}_{\text{Focal E-IoU}} = IOU^{\gamma}\mathcal{L}_{\text{EIoU}} \tag{20}$$

$$IOU = |A \cap B|/|A \cup B| \tag{21}$$

The loss value increases proportionally with the *IOU*, resulting in a larger loss for high-quality regression targets. This adjustment aids in enhancing accuracy, and $\gamma$ denotes the gradient's contribution. By adjusting $\gamma$, the algorithm assigns a greater gradient contribution to high-quality samples, thereby facilitating improved convergence of the function.

## 4. Experimental Results and Analysis

### 4.1. Experimental Environment and Evaluation Indicators

4.1.1. Experimental Environment

In order to ensure the effectiveness of the algorithm, the datasets utilized in this study were obtained exclusively from the UAS of Yunnan Power Supply Company. A total of 1627 images depicting insulators and various defects were selected for analysis. These defects encompassed four distinct types: insulator breakage (denoted as jyzps), insulator self-explosion (denoted as jyzzb), insulator fouling (denoted as jyzwh), and insulator binding wire loosening (denoted as jyzzxst). Table 1 lists the types of insulator defects and the number of training and validation sets corresponding to specific defect names. Figure 7 shows the images of different defect types (the images have been enlarged). For training purposes, the smallest YOLOv5s model was employed, with a batch size of 64 and a training batch size of 300. The laboratory setup environment is detailed in Table 2.

**Table 1.** Dataset type and quantity.

| Insulator Defect Type | Type of Label | Training Set | Detection Set |
|---|---|---|---|
| Insulator contamination | jyzwh | 269 | 50 |
| Insulator self-explosion | jyzzb | 726 | 140 |
| Tie the thread pine | jyzzxst | 234 | 50 |
| Broken insulator | Jyzps | 404 | 60 |

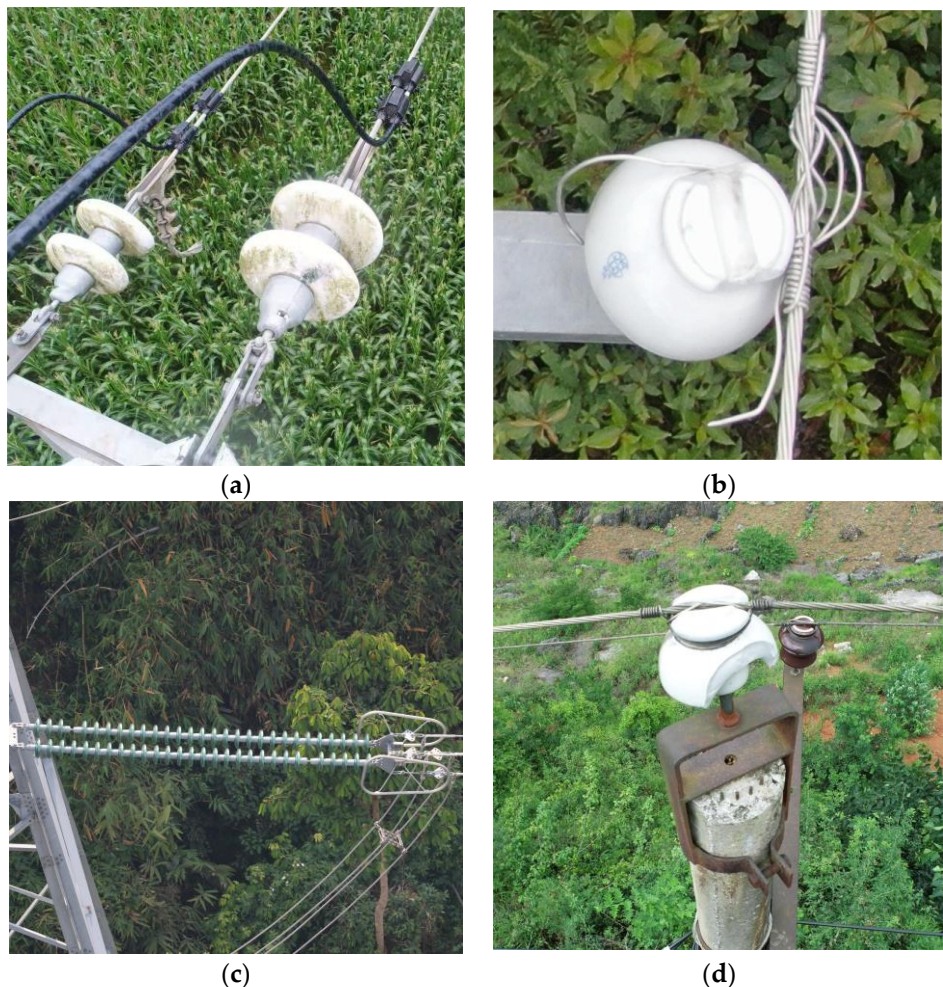

**Figure 7.** Pictures of insulator defects: (**a**) Insulator fouling; (**b**) loosening of insulator tie lines; (**c**) spontaneous detonation of insulator; (**d**) broken insulator.

**Table 2.** Experimental platform environment configuration.

| Environmental Configuration | Parameter |
| --- | --- |
| Operating system | Ubuntu 16.04 |
| GPU | Tesla T4 (16 GB) |
| CPU | Intel(R) Xeon(R) Processor v8 |
| Deep learning model framework | Pytorch 1.7.1 |
| GPU acceleration environment | CUDA 11.0.2 |
| Programming language | Python 3.8 |

4.1.2. Evaluation Index

In this paper, an evaluation of each algorithm filtering performance is conducted using two metrics: normalized mean square error (NMSE) and peak signal-to-noise ratio (PSNR). A higher PSNR value and a lower NMSE value indicate a superior filtering effect. The formulas for calculating these metrics are presented in Equations (22) and (23):

$$NMSE = \frac{\sum\limits_{i=0}^{M-1} \sum\limits_{j=0}^{N-1} [y(i,j) - I(i,j)]^2}{\sum\limits_{i=0}^{M-1} \sum\limits_{j=0}^{N-1} [I(i,j)]^2} \tag{22}$$

$$PSNR = 10 \cdot \lg\left(\frac{M \times N \times 255^2}{\sum\limits_{j=1}^{N}\sum\limits_{i=1}^{M}[y(i,j) - I(i,j)]}\right) \tag{23}$$

where $I(i,j)$ and $y(i,j)$ represent the gray values of the corresponding pixel points of the original image $I$ and the filtered output image $Y$, respectively, and $M$ and $N$ denote the length and width of the image, respectively.

In various application scenarios, diverse effects can be observed when employing distinct target test algorithms. The algorithm's reasoning performance primarily relies on several parameters, including the number of detection frames per second (FPS), the number of floating-point calculations (Flops), as well as the metrics of precision (P), recall (R), average accuracy (AP), and mAP. These metrics are denoted as (24), (25), (26) and (27) in the calculation representation:

$$P = \frac{TP}{TP + FP} \tag{24}$$

$$R = \frac{TP}{TP + FN} \tag{25}$$

$$AP = \int_0^1 P(R)dR \times 100\% \tag{26}$$

$$mAP = \frac{\sum\limits_{i=1}^{k} AP_i}{k} \tag{27}$$

where TP represents the correct number of targets, FP denotes the number of misunderstandings as the target, FN indicates the number of misunderstandings as non-targets, and k refers to the total number of targets. Generally, P and R are presented as the opposite performance trend, which is used as mAP to represent the comprehensive performance of target detection.

*4.2. Lutium Comparison*

4.2.1. Add the Effect of the Noise Reduction Module

To verify the accuracy of target detection, detection accuracy, and comprehensive performance indicators in drone shooting scenarios, the influence of detection accuracy and comprehensive performance indicators was examined. To investigate the impact of noise on the test results, different levels of noise density (0.2, 0.4, 0.6, and 0.8 for salt and pepper) were introduced into each dataset. The presence of noise and its effect on the test results are presented in Table 3.

**Table 3.** The effect of different levels of noise on the detection performance of YOLOv5.

| Noise Density | (All) P | (All) R | (All) mAp@0.5 | (All) mAp@0.5:0.95 |
|---|---|---|---|---|
| 0 | 0.734 | 0.653 | 0.706 | 0.487 |
| 0.2 | 0.719 | 0.636 | 0.666 | 0.472 |
| 0.4 | 0.711 | 0.632 | 0.665 | 0.469 |
| 0.6 | 0.705 | 0.571 | 0.646 | 0.468 |
| 0.8 | 0.703 | 0.603 | 0.64 | 0.462 |

Through ablation experiments, it can be deduced that noise exerts a substantial influence on the detection accuracy of YOLOv5. With each increment of 0.2 in noise density, the overall accuracy of defect detection experiences a decline from approximately 1% to 2%. Simultaneously, the mAP decreases by 0.02. The impact on the detection performance becomes more pronounced as the noise density increases.

To quantitatively assess the filtering effectiveness of the NW-AMF algorithm, ablation experiments were conducted by comparing it with other widely used industrial filtering algorithms, namely tri-state median filter (TMF) [38], switching median filter (SMF) [39], and adaptive center-weighted median filter (ACWMF) [40].The experimental results are shown in Table 4 and Figures 8 and 9.

**Table 4.** Filtering effects of different algorithms.

| Arithmetic | Norm | Noise Intensity | | | |
|---|---|---|---|---|---|
| | | **0.2** | **0.4** | **0.6** | **0.8** |
| TMF | PSNR | 28.82 | 25.45 | 18.50 | 10.38 |
| | NMSE | 0.0048 | 0.0105 | 0.0519 | 0.3373 |
| ACWMF | PSNR | 35.27 | 31.34 | 26.26 | 13.91 |
| | NMSE | 0.00011 | 0.0027 | 0.0087 | 0.1496 |
| SMF | PSNR | 35.89 | 31.95 | 27.60 | 16.32 |
| | NMSE | 0.0009 | 0.0023 | 0.0064 | 0.0858 |
| NW-AMF | PSNR | 39.34 | 34.86 | 31.76 | 28.44 |
| | NMSE | 0.0004 | 0.0012 | 0.0025 | 0.0053 |

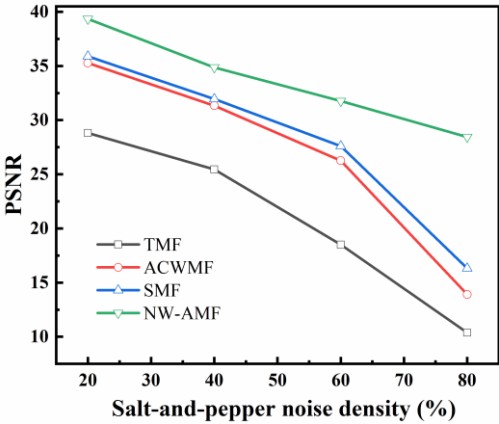

**Figure 8.** PSNR values for different algorithms.

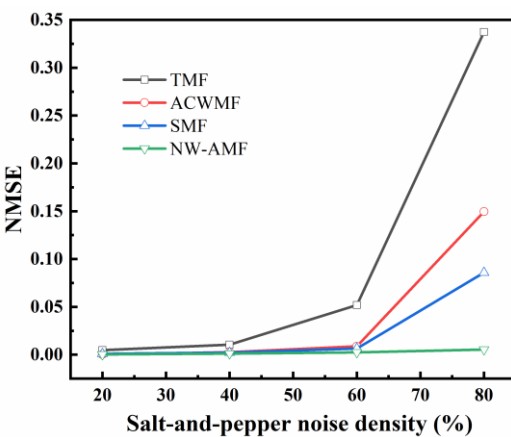

**Figure 9.** NMSE values for different algorithms.

Based on the graphical data, it is evident that the NW-AMF algorithm attains superior PSNR values and lower NMSE values when confronted with pretzel noise density values of 0.2, 0.4, 0.6, and 0.8. This observation signifies the algorithm's heightened filtering performance. Notably, as the noise density gradient increases, the performance indices of the SMF algorithm, ACWMF algorithm, and TMF algorithm undergo significant changes,

indicating a notable decline in their filtering performance. Conversely, the NW-AMF algorithm exhibits a more gradual alteration in the aforementioned indices, thereby maintaining a superior performance index even under the influence of 0.8 density salt-and-pepper noise. Consequently, the algorithm proposed in this study demonstrates a consistent and stable ability to remove salt-and-pepper noise of varying densities. Hence, it can be inferred that the algorithm presented in this paper possesses a reliable capacity to eliminate pretzel noise of different densities.

The resultant effect of the noise reduction module on the noisy image is presented in Table 5.

**Table 5.** YOLOv5 Detection effect after noise reduction for pictures with different noise density.

| Noise Reduction Level | (All) P | (All) R | (All) mAp@0.5 | (All) mAp@0.5:0.95 |
|---|---|---|---|---|
| 0.2 | 0.730 | 0.654 | 0.701 | 0.472 |
| 0.4 | 0.727 | 0.653 | 0.691 | 0.472 |
| 0.6 | 0.723 | 0.655 | 0.688 | 0.471 |
| 0.8 | 0.719 | 0.651 | 0.694 | 0.47 |

The experimental findings from Table 5 provide substantial evidence that the noise reduction module has significantly enhanced image detection performance. Following the application of the noise reduction module, there is an average increase of approximately 2% in accuracy, accompanied by a marginal improvement of 0.01 in the comprehensive performance mAP.

To validate the non-end-to-end detection effect of the joint noise reduction and target detection network, an ablation experiment was conducted using a dataset containing pretzel noise with a density of 0.4. This dataset had already undergone the noise reduction process to assess the impact on various target detection algorithms. The outcomes of this experiment are presented in Table 6.

**Table 6.** Effectiveness of different algorithms for noise reduction image detection.

| Method | (All) P | (All) R | (All) mAp@0.5 | (All) mAp@0.5:0.95 |
|---|---|---|---|---|
| YOLOv3 | 0.701 | 0.638 | 0.634 | 0.414 |
| YOLOv5s | 0.727 | 0.653 | 0.691 | 0.472 |
| SSD-VGG | 0.612 | 0.562 | 0.513 | 0.312 |
| YOLOv5-RepVGG | 0.691 | 0.591 | 0.603 | 0.402 |
| YOLOv7 | 0.731 | 0.647 | 0.693 | 0.47 |
| YOLOv8 | 0.739 | 0.65 | 0.701 | 0.488 |
| RcpVGG-Yolov5 | 0.749 | 0.659 | 0.705 | 0.503 |

By comparing the algorithm proposed in this paper with both the widely adopted industry-standard target algorithms and the current state-of-the-art approaches, it can be concluded that the non-end-to-end nature of the algorithm does not adversely affect the accuracy of the detection results or the performance metrics such as mAP.

### 4.2.2. Improvement of the Loss Function

Prior to evaluating the impact of the Focal EIOU loss function, it is necessary to fine-tune the parameter $\gamma$ to align it more closely with the dataset, thereby facilitating enhanced convergence. The ablation experiments, as depicted in Table 7, serve to substantiate this claim.

Drawing upon the insights from the table, it can be concluded that achieving an equilibrium between the positive and negative samples to the training when $\gamma = 0.5$ is a more appropriate approach. This is particularly pertinent when considering the limited size of the target dataset in this experiment.

To validate the efficacy of the loss function, the experiments were conducted in accordance with the prevailing trend of conducting ablation experiments on loss functions.

YOLOv5-6.0, which employs the CIOU loss function, served as the baseline for comparison. Consequently, the loss function was substituted with EIOU, Focal EIOU, Focal WIOU, and SIOU. The outcomes of these experimental iterations are presented in Table 8.

**Table 7.** Detection performance for different parameter values.

| $\gamma$ Parameter Value | (All) P | (All) R | (All) mAp@0.5 | (All) mAp@0.5:0.95 |
|---|---|---|---|---|
| 0.1 | 0.762 | 0.672 | 0.717 | 0.498 |
| 0.3 | 0.779 | 0.665 | 0.722 | 0.499 |
| 0.5 | 0.788 | 0.631 | 0.738 | 0.502 |
| 0.7 | 0.781 | 0.639 | 0.731 | 0.502 |
| 0.9 | 0.774 | 0.655 | 0.729 | 0.503 |

**Table 8.** Different loss functions detection renderings.

| Loss Function Type | (All) P | (All) R | (All) mAp@0.5 | (All) mAp@0.5:0.95 |
|---|---|---|---|---|
| CIOU (initial) | 0.754 | 0.653 | 0.706 | 0.497 |
| EIOU | 0.7 | 0.681 | 0.691 | 0.495 |
| SIOU | 0.762 | 0.656 | 0.722 | 0.499 |
| Focal WIOU | 0.768 | 0.642 | 0.738 | 0.502 |
| Focal EIOU | 0.788 | 0.63 | 0.746 | 0.503 |

Upon observing Figures 10 and 11 as well as Table 8, it is evident that the YOLOv5s loss functions, namely CIOU and SIOU, exhibit a tendency towards stable detection accuracy and convergence speed. However, these loss functions demonstrate subpar accuracy when confronted with images containing severe imbalances between positive and negative samples, and their convergence speed is deemed inadequate. Conversely, a closer analysis of the chart reveals that the algorithm's accuracy is enhanced by approximately 2% when combined with the Focal loss function, resulting in an overall improvement of approximately 0.05 in comprehensive performance. Notably, the convergence effect of the analysis diagram indicates that the Focal EIOU loss function exhibits faster and more effective convergence.

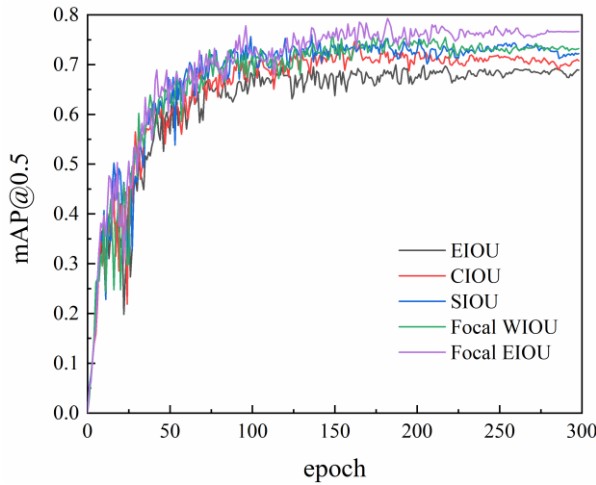

**Figure 10.** mAP curve in the training process.

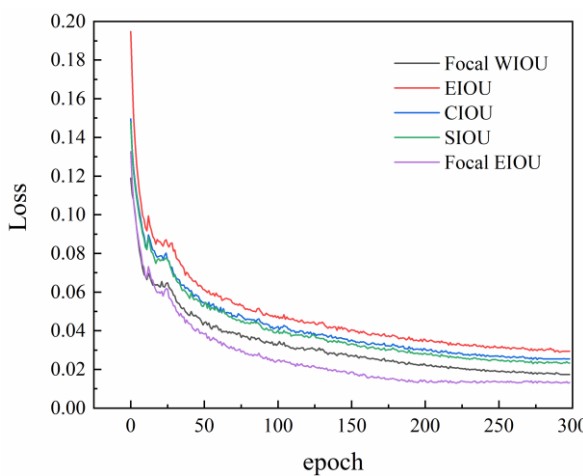

**Figure 11.** The loss curve in the training process.

4.2.3. Improvement of the Effect of the Main Network of RepVGG

To verify the effectiveness of the RcpVGG-YOLOv5 algorithm following enhancements to the primary network, a detection contrast analysis is conducted using an image density of 0.4. The detection performances of YOLOv5s [20], YOLOv5-RepVGG [25], YOLOv3 [19], SSD-VGG [17], YOLOv7 [41], and YOLOv8 [42] are compared, and the results are presented in Table 9.

**Table 9.** Target detection performance comparison.

| Method | (All) P | (All) R | (All) mAp@0.5 | FPS (Hz) |
|---|---|---|---|---|
| YOLOv3 | 0.691 | 0.635 | 0.632 | 109 |
| YOLOv5s | 0.712 | 0.636 | 0.646 | 139 |
| SSD-VGG | 0.605 | 0.559 | 0.58 | 203 |
| YOLOv5-RepVGG | 0.682 | 0.592 | 0.59 | 218 |
| YOLOv7 | 0.722 | 0.649 | 0.676 | 155 |
| YOLOv8 | 0.734 | 0.652 | 0.681 | 183 |
| RcpVGG-YOLOv5 | 0.749 | 0.659 | 0.705 | 207 |

Figure 12 provides a clear visualization of the superior comprehensive performance (mAP) exhibited by the RcpVGG-YOLOv5 algorithm compared to the other algorithms. Furthermore, Figures 13 and 14 demonstrate that RcpVGG-YOLOv5 achieves the highest accuracy and exhibits the most robust convergence under complex conditions. Ultimately, the superiority of RcpVGG-YOLOv5 is visually evident in Figure 15.

To offer a more intuitive representation of the performance impact, the following figures showcase the detection results of various algorithms on four distinct defective datasets: spontaneous detonation of insulator defect, broken insulator defect, loosening of tie lines insulator defect, and fouled insulator defect (Figure 16, Figure 17, Figure 18, and Figure 19, respectively) (the detection results of YOLOv7, YOLOv8, and the algorithm proposed in this paper do not exhibit significant differences, and therefore, this aspect of the results has been omitted).

In the context of self-explosive defects, the recognition rate of insulators in the VGG/RepVGG network tends to be low, thereby increasing the likelihood of missed judgments. While YOLOv5s demonstrates the capability to detect defects, it is prone to misinterpret obscuration as a dirty defect on the insulator. Similarly, YOLOV3 can only marginally identify the defect, but it often confuses the end and tail of the glass insulator with a defect. Conversely, RcpVGG-YOLOv5 exhibits adaptability in detecting scene defects.

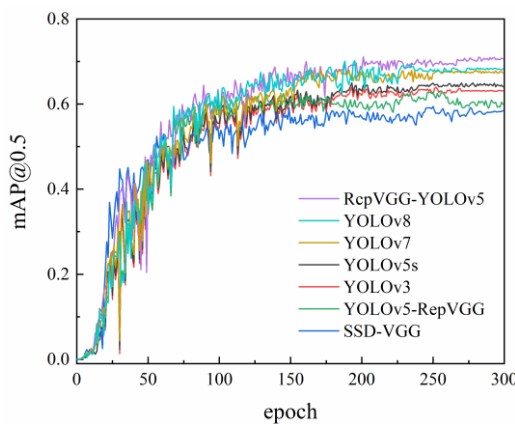

**Figure 12.** mAP of various algorithms.

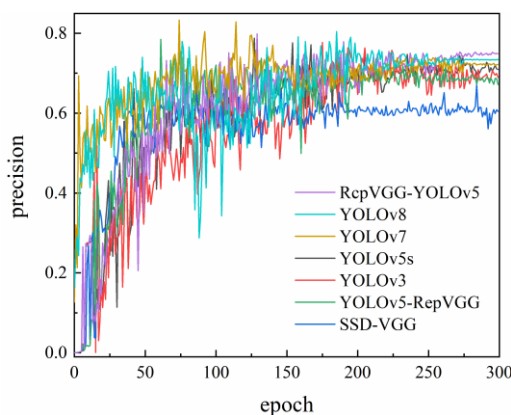

**Figure 13.** The accuracy of various algorithms.

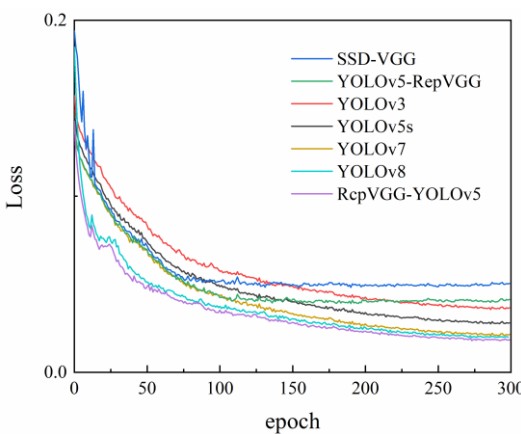

**Figure 14.** The loss value of each type of algorithm.

Regarding damage defects in insulators, particularly in the presence of salt and pepper noise, on the one hand, the VGG/RepVGG network exhibits a low recognition rate for small target insulation areas. Moreover, it is prone to erroneously identifying non-target areas as targets. While YOLOv5s demonstrates the ability to detect defects, it is susceptible to noise interference, leading to erroneous recognition of insulator dirt defects. YOLOv3 also suffers from recognition errors. On the other hand, RcpVGG-YOLOv5 showcases adaptability in scene defect detection.

With respect to recognizing defects in multiple insulators, the VGG/RepVGG network is significantly impacted by noise, resulting in a low recognition rate. Both YOLOv5s and

YOLOv3 can only identify the absence of prominent ligation lines, which increases the likelihood of overlooking loose defects during inspections. However, RcpVGG-YOLOv5 demonstrates adaptability in detecting defects within a scene.

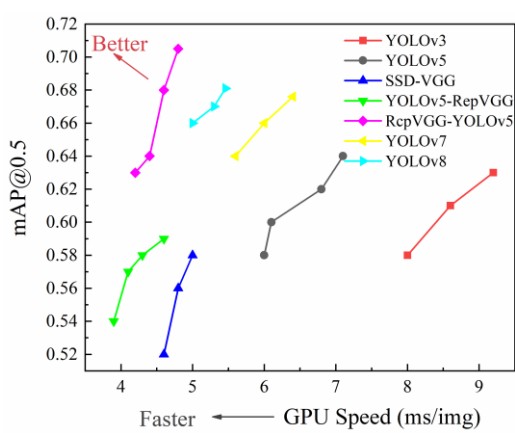

**Figure 15.** Inference speed.

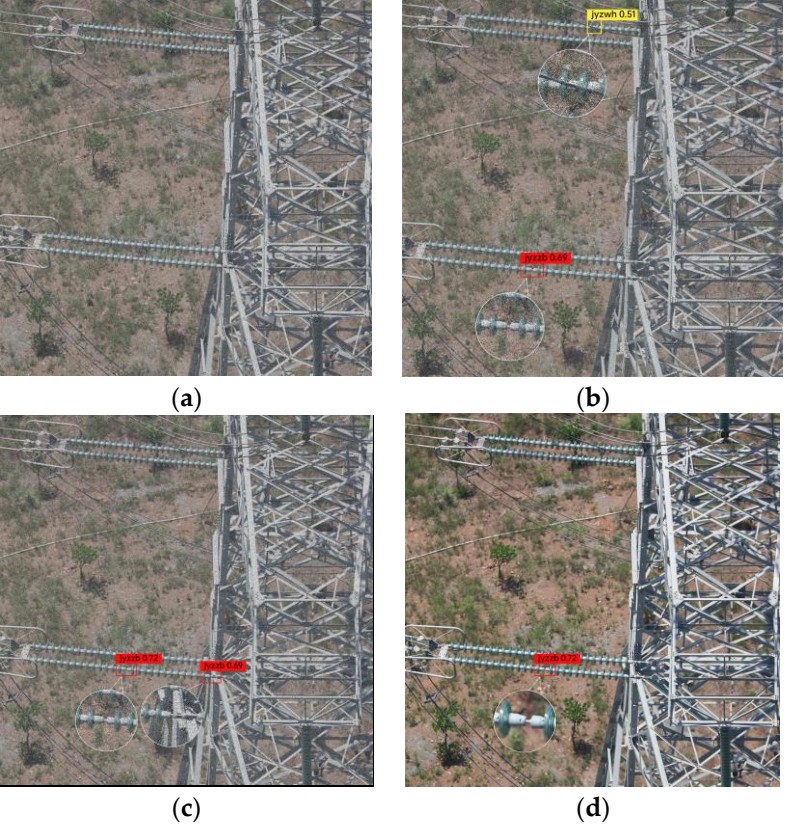

**Figure 16.** Spontaneous detonation of insulator defect: (**a**)VGG/RepVGG; (**b**) YOLOv5s; (**c**) YOLOv3; (**d**) RcpVGG-YOLOv5.

In the context of recognizing multiple insulators, the VGG/RepVGG network exhibits a significant issue of missed inspections. Both YOLOv5s and YOLOv3 are susceptible to misinterpretations when inspecting the loosening of tie lines using the side binding method. However, RcpVGG-YOLOv5 demonstrates adaptability in detecting defects within a scene.

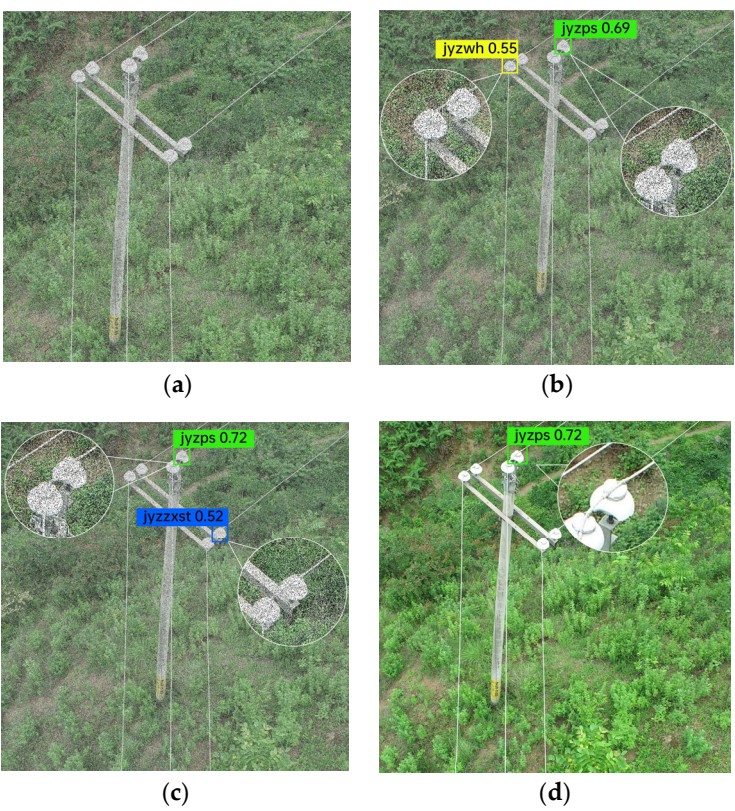

**Figure 17.** Broken insulator defect: (**a**) VGG/RepVGG; (**b**) YOLOv5s; (**c**) YOLOv3; (**d**) RcpVGG-YOLOv5.

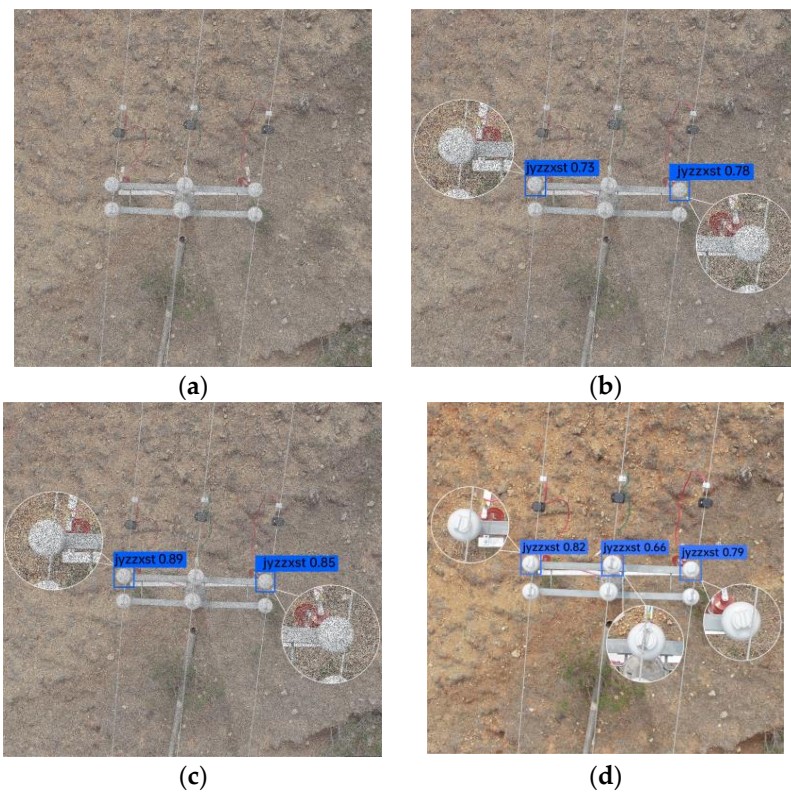

**Figure 18.** Loosening of insulator tie lines defect: (**a**) VGG/RepVGG; (**b**) YOLOv5s; (**c**) YOLOv3; (**d**) RcpVGG-YOLOv5.

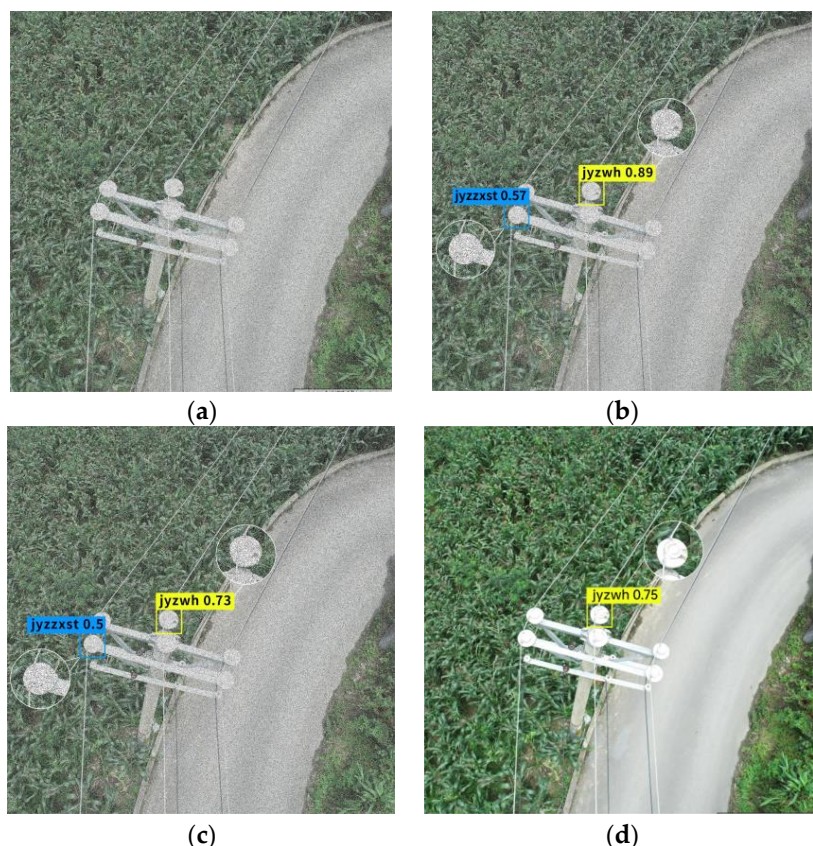

**Figure 19.** Fouled insulator defect: (**a**) VGG/RepVGG; (**b**) YOLOv5s; (**c**) YOLOv3; (**d**) RcpVGG-YOLOv5.

Based on the empirical findings, it has been observed that the SSD-VGG and YOLOv5-RepVGG algorithms exhibit a suboptimal performance in recognizing small- and medium-sized targets in machine patrol images. Moreover, the inclusion of salt and pepper noise with a density of 0.4 further exacerbates the inspection rate. Consequently, direct implementation of these algorithms in practical engineering applications is deemed unsuitable. Conversely, the YOLOv3 and YOLOv5s algorithms demonstrate the capability to identify insulators; however, when subjected to the aforementioned interference of salt and pepper noise with a density of 0.4, the rates of missed inspections and misinterpretations increased. In light of these observations, the RcpVGG-YOLOv5 algorithm emerges as a more viable solution, exhibiting enhanced adaptability to such complex scenarios.

## 5. Discussion

According to the analysis of experimental results, the enhanced algorithm network proposed in this study (0.705) exhibits superior comprehensive performance (mAP) compared to the baseline network YOLOv5-RepVGG (0.59) and the base algorithm YOLOv5s (0.646). Furthermore, it outperforms traditional algorithms such as SSD-VGG (0.58) and YOLOv3 (0.632). Notably, when compared to the latest algorithms, it demonstrates a performance advantage of 2.9% over YOLOv7 and 2.4% over YOLOv8, thereby establishing the algorithm's superiority in terms of performance. In terms of accuracy, it also surpasses the baseline algorithm (0.712) and the baseline network (0.682), exhibiting a clear advantage with accuracy rates 2.7% and 1.5% higher than the latest algorithms YOLOv7 and YOLOv8, respectively. Regarding speed, the proposed algorithm demonstrates a notable advantage, with a 33.5% increase compared to YOLOv7 and a 13.1% increase compared to YOLOv8. This is visually depicted in Figure 15, which clearly illustrates the superior balance between accuracy and speed achieved by the algorithm presented in this paper. Furthermore, in terms of effectiveness, the algorithm proposed in this study exhibits fewer instances of

misdetection and leakage in the detection of 300 insulator defect maps compared to other algorithms. This characteristic renders it more suitable for engineering applications.

## 6. Conclusions

In this paper, we introduce an algorithm, namely RcpVGG-YOLOv5, which combines noise reduction and target detection techniques. The experimental findings demonstrate that the enhanced algorithm remains effective and exhibits robustness when dealing with aerial images amidst complex background conditions. In practical scenarios, this algorithm facilitates prompt response by maintenance personnel to defects, thereby offering technical assistance for line detection in power supply companies. It is important to acknowledge that our model does encounter certain limitations. Firstly, the algorithm network lacks a detection strategy for insulators, such as small targets, apart from the loss function component. This aspect presents ample room for improvement. Secondly, the overall dataset size is relatively small and necessitates supplementation. Consequently, the accuracy rate does not surpass 80%. Nevertheless, the model proposed in this study retains high practicality and supports the development of applications in conjunction with the front-end, thereby enabling easy generalization.

Henceforth, future work will focus on two primary areas. Initially, the restricted availability of the transmission line defect dataset due to confidentiality concerns has limited the sample collection process. Consequently, efforts will be made to continue gathering insulator defect samples with varying scales and backgrounds, thereby expanding the experimental dataset and enhancing the detection model's generalization capabilities. Furthermore, the network structure will undergo further optimization, incorporating targeted strategies for small target detection. This optimization will result in an improved detection performance, enabling real-time and efficient identification of transmission line defects.

**Author Contributions:** Conceptualization, F.S.; Methodology, Z.Y. and Y.L.; Software, Y.L.; Validation, Y.L.; Resources, S.Z.; Data curation, Y.L.; Writing—original draft, Y.L.; Writing—review & editing, Z.Y., F.S. and Y.Y.; Supervision, F.S. and Y.Y.; Project administration, S.Z.; Funding acquisition, Z.Y. All authors have read and agreed to the published version of the manuscript.

**Funding:** This work was not supported by project funding.

**Data Availability Statement:** Due to the confidentiality of the dataset for this study, participants in this study did not agree to share their data publicly, and therefore supporting data could not be provided.

**Conflicts of Interest:** The authors declare no competing interests.

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
