# Peer review of "Research on Identification and Detection of Transmission Line Insulator Defects Based on a Lightweight YOLOv5 Network"

_remotesensing, doi:10.3390/rs15184552_

Round 1

Reviewer 1 Report

This paper proposes a method for detection and recognition of transmission lines. Compared to previous works, for example YOLOv5, this paper mainly focuses on adding a denoising filter (NW-AMF) before detecting and lightweight network (RepVGG) for more feasibility. It has novel application scenario and certain application prospects. However, I have a few comments below.

1. The proposed method is not an end-to-end network. It's commonly known that end-to-end network can improve the effectiveness of deep-learning network. The author should explain in this task of detecting transmission lines or similar works, whether adopting un-end-to-end network would affect the effectiveness or not. For example, you can cite some well-known works which is relevant to your topic.

2. The framework could be improved, for the current framework is somewhat confusing. For example, '1.YOLOv5 network' should be a common knowledge within the detecting community, thus should be summarized less. Besides, I think '1.2YOLOv5 loss function' should be better removed to loss function section 2.3. Also, as a background knowledge, it should be summarized less, because it's NOT your contribution.

3. The authors claim that the loss function is novel and improved. In this section, more relevant works should be cited to reinforce your opinion. Because the key of loss function lies in hyperparameters configuration, while in your loss function, all weight is equal (as 1). It should be discussed despite some improvements have been achieved.

4. The NW-filtering proposed by the authors in section 2.1 lacks innovation, and I personally think that denoising is just a usual pre-processing method, which should not be taken as the core theoretical part of the article. Even if the author's denoising method is very good, it should be compared with the state-of-the-art methods.

Minor issues:

1. There are many errors regarding the format issues. For example, number should be superscript within author list in page 1 line 3. Besides, the best results should be highlighted in bold to verify your superiority in each comparison table. What's more, almost all fonts by mathtype is not consistent with word font. It should be revised to be consistent.

2.  I think demonstrating Figure 1 may not be appropriate. You should give a more vivid network illustrating your network, from start to end, rather than part of it. It's important as the beginning of an article.

3.  What does each item in the "type of label" column in Table 1 mean? Could the authors please label or explain accordingly?

4. EIOU Loss does not cite the original author's article.

Reviewer 2 Report

-

Reviewer 3 Report

First of all, I would like to congratulate the authors for the developed research.

1. Figure 1 should be resized, for the caption to be included on the same page.

2. It is not clear what amount of data was used for training and what amount for testing.

3. Figures 13 to 16 should be enlarged. Maybe it is a better idea to stack them in two rows.

4. Equation numbering should be aligned.

5. Equation font should be the same size as the rest of the text.

English language is OK.

Reviewer 4 Report

Detecting the defect of transmission line based on UAV view is studied in this paper. To ensure the quality of the image, a denoising preprocessing is applied before the detection by an adaptive neighborhood weighted median filter (NW-AMF). Defect detection is carried out by a modified YOLOv5 in which RcpVGG is utilized to replace the original backbone. Focal Loss and EIOU loss are combined as the loss function of the modified YOLOv5. These operations are expected to achieve the balance between the accuracy and speed. Unfortunately, the writing of this paper is unprofessional and many sentences are rough, leading to difficulty to read.

Questions:

1. There is a big problem in sentence writing, and a large number of sentences are rough.

2. Pay more attention to the details of the format. There is a duplicate formula in section 1.2

3. The combination of Focal Loss and EIOU Loss in formula (20) is not clear. Further interpretation should be required.

4. There are many efficient and lightweight backbone networks, so why choose RepVGG? The modification is to remove the batch normalization layer and the 1x1 convolution layer. What is the purpose?

5. In the comparative experiments, the comparison with yolov7 and yolov8 should be required.

6. The figures in the experiments are not high-definition. And there should be a figure to display the results of the VGG-YOLOv5 ?

The writing of this paper is unprofessional and many sentences are rough, leading to difficulty to read.

Round 2

Reviewer 1 Report

The authors have addressed the majority of my concerns by revising the manuscript based on my comments, so it should be accepted. 

Besides, please modify format to be confirmed with Remote Sensing publish requirements. Minor suggestions: 

1. I think introduction section should be titled with section 1, while the methodology section should be started with title 'section 2'. Please confirm with the template of this journal. If so, please modify it. 

2. As now, the reference works are not highly related with Remote Sensing. Most references cited in this manuscript are from computer vision community. Hence, I encourage the authors should cite more relevant works, especially from Remote Sensing. For example, DOI: 10.3390/rs14143421; 10.3390/rs13020230; 10.3390/rs14133229; 10.3390/rs15051412. 

Author Response

Point 1:I think introduction section should be titled with section 1, while the methodology section should be started with title 'section 2'. Please confirm with the template of this journal. If so, please modify it.

Response 1:Thank you very much for your formatting questions about the manuscript, which has been corrected according to the template for title numbering and labeled in red.

Point 2:As now, the reference works are not highly related with Remote Sensing. Most references cited in this manuscript are from computer vision community. Hence, I encourage the authors should cite more relevant works, especially from Remote Sensing. 

Response 2:Your questions about the manuscript references are much appreciated. We have added to the references as you requested. This time, we have added six references in remote sensing, numbered 13-18 (Introduction to this paper, line 60, line 66, and lines 89-94).

Reviewer 3 Report

The authors made the necessary improvements to the revised manuscript can be published.

Author Response

再次感谢您对手稿的仔细检查。